# Development of the National Institute of Health Healing Experience of All Life Stressors Short Form (NIH-HEALS-SF)

**Marcelli Cristine Vocci**[1], **Polycarpe Bagereka**[1], **Rezvan Ameli**[1], **Ninet Sinaii**[2], **Jeremy L. Davis**[3], **Manish Agrawal**[4], **Ann Berger**[1]*

1 Pain and Palliative Care Service, Clinical Center, National Institutes of Health, Bethesda, Maryland, United States of America, 2 Biostatistics and Clinical Epidemiology Service, Clinical Center, National Institutes of Health, Bethesda, Maryland, United States of America, 3 Surgical Oncology Program, National Cancer Institute, National Institutes of Health, Bethesda, Maryland, United States of America, 4 Sunstone Therapies, Rockville, Maryland, United States of America

* aberger0123@gmail.com

## Abstract

The National Institutes of Health Healing Experience of All Life Stressors (NIH-HEALS) is a validated measure of psychosocial-spiritual well-being with strong psychometric properties, supporting its use in both research and clinical settings. To enhance its applicability in large-scale studies and routine clinical practice while minimizing patient burden, a short-form version, the NIH-HEALS-SF, was developed. Using data from 200 participants, Least Absolute Shrinkage and Selection Operator (LASSO) regression was applied to identify the most predictive and relevant items from the original scale. Each item was assessed within three core domains: Connection, Reflection and Introspection, and Trust and Acceptance. The final nine-item version was validated in an independent sample of 164 individuals from three distinct cohorts. Psychometric evaluation demonstrated strong internal consistency (Cronbach's alpha: 0.75–0.85) and high correlation with the original full-scale NIH-HEALS (rp = 0.92–0.96). These findings suggest that the NIH-HEALS-SF maintains the conceptual integrity and measurement properties of the original instrument while reducing administration time. By offering a concise yet robust assessment of psychosocial-spiritual well-being, the NIH-HEALS-SF may facilitate broader implementation in both clinical and research contexts, particularly in settings where time-efficient assessments are needed.

## Introduction

Recent decades have seen an increased interest in global assessments of well-being, which reflects an increased recognition of the importance of psychosocial-spiritual well-being in health-care [1–4]. Indeed, the World Health Organization (WHO) considers health a state of complete physical, mental, and social well-being and not merely the absence of disease or infirmity [5]. To identify psychosocial-spiritual healing and interventions that may help individuals the Healing Experience in All Life Stressors (NIH-HEALS) [6,7] was developed. The measurement tool was validated at the National Institutes of Health (NIH), and focused

the Creative Commons CC0 public domain dedication.

**Data availability statement:** Anonymized raw data set from the four samples used in our study, which are owned by the authors, to Zenodo: https://doi.org/10.5281/zenodo.14806363.

**Funding:** This study was funded by the Intramural Program of the NIH Clinical Center. Funding was provided by the National Institutes of Health (NIH) to AB. The funders had no role in study design, data collection and analysis, decision to publish, or preparation of the manuscript.

**Competing interests:** The authors have declared that no competing interests exist.

on assessing an individual's psychosocial-spiritual wellbeing among patients with life-limiting illnesses. This questionnaire consists of 35-items with three factors (Connection, Reflection & Introspection, and Trust & Acceptance).

NIH-HEALS is simple to use and has been applied in several contexts and populations, including adult patients with serious and/or debilitating diseases [6–9], women infected with the human immunodeficiency virus (HIV) [10], and advanced cancer patients [4,11]. The NIH-HEALS has also been used in an intervention trial using psilocybin in patients with advanced cancer and has shown sensitivity to change [4]. Furthermore, the NIH-HEALS has been adapted for use in Uganda [11], suggesting it can be adapted for use in different cultures.

Although the 35-item version of NIH-HEALS has been successfully used, a shorter version is desirable. Indeed, a short-form of any instrument is considered to be more user friendly, and enhances its applicability particularly in clinical settings and large-scale studies, and reduces patient burden [12,13].

The current study describes the procedures involved in the development of the NIH HEALS short form, which utilized a pre-existing dataset to identify the optimal items and three independent samples to conduct a comprehensive analysis of its psychometric properties.

## Methods

### Study design

This study employs a scale reduction methodological approach in developing the short form of the NIH-HEALS questionnaire, utilizing four independent samples. First, a pre-existing sample of 200 patients with serious and/or life-limiting illnesses was used to identify the items that would comprise the short-form version (Table 1). Further details about this sample can be found elsewhere [7]. For the psychometric analysis, the nine most robust items were selected for the short form and were applied to 164 subjects from three different samples: Sample 2 consisted of COVID-19 healthcare workers; Sample 3 comprised patients and family members

**Table 1. Datasets used for the development and psychometric analysis.**

| | | Title | Subjects | Context | IRB* approval |
|---|---|---|---|---|---|
| Development-Item selection | Sample 1 | The National Institutes of Health Measure of Healing Experience of All Life Stressors (NIH-HEALS): Factor Analysis and Validation [7] | 200 | Patients with severe and/or life-threatening diseases were recruited for the factor analysis and validation of the NIH-HEALS. | National Institutes of Health Institutional Review Board NCT02664402 |
| Psychometric analysis | Sample 2 | The Effect of a Combined Nature-Based and Audio-Based Mindfulness Intervention on Stress Among Frontline COVID-19 Healthcare Workers | 78 | COVID-19 healthcare workers were recruited to assess the feasibility and effectiveness of a combined Nature and Audio-based Mindfulness stress reduction program. | National Institutes of Health Institutional Review Board NCT04846790 |
| | Sample 3 | Hereditary Gastric Cancer Syndromes: An Integrated Genomic and Clinicopathologic Study of the Predisposition to Gastric Cancer | 56 | Patients and family members, who fulfilled clinical criteria for a hereditary gastric cancer syndrome irrespective of previous genetic testing or treatment. | National Institutes of Health Institutional Review Board NCT03030404 |
| | Sample 4 | Psilocybin-Assisted Therapy Improves Psycho-Social-Spiritual Well-Being in Cancer Patients [4] | 30 | Cancer patients with major depressive disorder to assess changes in NIH-HEALS scores as they underwent psilocybin-assisted therapy. | Advarra Institutional Review Board NCT04593563 |

*IRB, Institutional Review Board.

who met clinical criteria for hereditary gastric cancer; and Sample 4 consisted of patients diagnosed with cancer (Table 1). These independent samples were participants in individual studies that utilized their own sampling techniques for participant selection; however, these studies were selected because they represented the target population with varying degrees and types of serious and/or life-limiting illnesses and chronic stress.

## Subjects

This study included a total of 364 participants. Sample 1 consisted of 200 patients with severe and/or life-threatening diseases, aged 18–89 years (median 52, IQR 34-59); 53% were female and most participants were Caucasian (72%). Sample 2 consisted of 78 healthcare workers who cared for patients with COVID-19, with a median age of 35 years (IQR 30-40) between 23-46; 86% identified as female and 73% were Caucasian. Sample 3 consisted of 56 adult patients and family members who met the clinical criteria for a hereditary gastric cancer syndrome, aged 18-70 years old with a median age of 40 years (IQR 33.3-50.5). Most participants were female (82%) and Caucasian (96%). Sample 4 consisted of 30 cancer patients with major depressive disorder aged 30-78 years old with a median age of 59 years (IQR 50-65); 70% identified as female and most participants were Caucasian (83%) (Table 2).

**Table 2. Demographic characteristics of included participants.**

| Characteristic | Total | Sample 1 | Sample 2 | Sample 3 | Sample 4 |
|---|---|---|---|---|---|
| | (n=364) | (n=200) | (n=78) | (n=56) | (n=30) |
| Age, in years* | 44.0 | 52.0 | 35.0 | 40.0 | 59.0 |
| Median (IQR) | (34.0-59.0) | (38.5-62.0) | (30.0-40.0) | (33.3-50.5) | (50.0-65.0) |
| Range (Min-Max) | 18-89 | 18-89 | 23-46 | 18-70 | 30-78 |
| Gender, n (%)** | | | | | |
| Male | 119 (33.3) | 90 (46.6) | 10 (12.8) | 10 (17.9) | 9 (30.0) |
| Female | 237 (66.4) | 103 (53.4) | 67 (85.9) | 46 (82.1) | 21 (70.0) |
| Race, n (%) α | | | | | |
| Asian | 27 (7.6) | 13 (6.8) | 12 (15.4) | 0 | 2 (6.9) |
| Black | 37 (10.4) | 30 (15.6) | 2 (2.6) | 2 (3.6) | 3 (10.3) |
| Caucasian | 274 (77.2) | 139 (72.4) | 57 (73.1) | 54 (96.4) | 24 (82.8) |
| All others/ Mixed | 17 (4.8) | 10 (5.2) | 7 (9.0) | 0 | 0 |
| Ethnicity, n (%) β | | | | | |
| Hispanic | 23 (6.6) | 13 (6.9) | 7 (9.2) | 2 (3.6) | 1 (3.3) |
| Non-Hispanic | 328 (93.5) | 176 (93.1) | 69 (90.8) | 54 (96.4) | 29 (96.7) |
| NIH-HEALS γ Scores, mean ± SD | | | | | |
| Total δ | 128.9 ±19.6 | 132.9 ±18.6 | 118.0 ±18.7 | 136.4 ±16.2 | 119.1 ±19.4 |
| Factor 1 | 35.2 ±10.5 | 37.7 ±9.8 | 28.9 ±10.6 | 36.3 ±9.4 | 30.8 ±9.4 |
| Factor 2 | 54.7 ±6.9 | 54.8 ±6.8 | 53.1 ±7.2 | 56.1 ±6.5 | 55.7 ±6.8 |
| Factor 3 | 39.3 ±7.7 | 40.4 ±6.7 | 35.6 ±7.0 | 43.9 ±7.0 | 32.6 ±8.0 |

IQR: inter-quartile (25th-75th percentile) range; SD: standard deviation.

*n=16 in Sample 1 were missing age data.

**n=1 in Sample 2 reported 'other' for gender.

α n=9 were missing race data.

β n=13 were missing ethnicity data.

γ NIH-HEALS: NIH Healing Experience in All Life Stressors [6,7].

δ n=15 missing scores due to missing responses to any question item.

## Instrument

The development and validation of the NIH-HEALS-35 is well documented [6,7]. The validation study with 200 subjects [7] resulted in a 35-item questionnaire. Scores are calculated using a 5-point Likert scale, with a range from Strongly Disagree to Strongly Agree. Its internal consistency was reported as Cronbach's α = 0.89, and split half-reliability = 0.95. Furthermore, its convergent ($r_s$ = 0.64, p < 0.0001) and divergent validity ($r_s$ = -0.34, p < 0.0001) were established by significant correlations between NIH-HEALS and its three factors with the Self Integration Scale (SIS)[14] and the Functional Assessment of Chronic Illness Therapy-Spiritual Well-Being (FACIT-Sp-12) [15].

## Statistical analyses

Medical records or archived samples were accessed for research purposes from April 2023 to March 2024. Results are reported as frequency (percentage), or using the median (interquartile range, IQR) and range (min-max), depending on the type of variable. Continuous data were assessed for normality, and appropriate statistical methods were used, as needed. Exploratory and confirmatory factor analyses were carried out during the validation phases of the NIH-HEALs 35-item and results are reported elsewhere [7]. For the purpose of scale reduction and development of short form in this study, Least Absolute Shrinkage and Selection Operator (LASSO) [16–18] regression modeling using a forward-selection method was carried out. LASSO enables the selection of the most important items from an instrument (variable selection), ensuring that the reduced version maintains its accuracy and effectiveness in capturing key information while simplifying the instrument's administration and interpretation (regularization). LASSO modeling works well with correlated items within the NIH-HEALS tool and has the capability of reducing variance. In addition, LASSO provided the advantages of a simpler tool that would be easier to use and interpret. For a sequence of regularization parameters, models were fit by maximizing the penalized log-likelihoods using the Schwarz Bayesian criterion (SBC) until its optimal value was reached for the selection of candidate questionnaire items. This process entailed the iterative addition of items for the entire NIH-HEALS, and separately for each of its factors (Table 3). These candidate items additionally underwent a qualitative review to confirm a sensible and clinically valid set of selections. To maintain the underlying constructs of the full scale, the results and rankings considered and proportionally weighted candidates for each factor. A previous sample of 200 patients (Sample 1)[7] was used for this development phase that led to the final 9-item short form.

The selected items were tested for reliability, where Cronbach's alpha and split-half reliability using the Spearman-Brown formula were used to ensure the items measure the same construct (internal consistency). The coefficient values closer to 1 indicated higher internal consistency among items, and values greater than 0.7 were considered acceptable. Pearson's correlation analysis was conducted to assess the relationship between the 9 items selected through LASSO analysis and the original 35-item version of the instrument. The objective of the Pearson's correlation was to determine the degree of linear association between responses to the short form and the full version, aiming to identify patterns of joint variation. Data were analyzed using SAS v9.4(proc glmselect, with selection=LASSO based on Schwarz Bayesian Criterion [choose=SBC]).

## Ethics statement

The study was conducted in accordance with the Declaration of Helsinki. This study was submitted and approved by the NIH Office of Human Subject Research Protection (OHSRP)

**Table 3. NIH-HEALS question items sorted by Least Absolute Shrinkage and Selection Operator (LASSO) regression model selections, for the entire set of questions and by factors.**

| Model Selection Iterative Step | Full Set of Question Items | NIH-HEALSα | | |
| --- | --- | --- | --- | --- |
| | | Factor 1 | Factor 2 | Factor 3 |
| 1 | q14* | q15* | q9* | q30* |
| 2 | q13* | q14* | q31* | q6 |
| 3 | q30* | q18 | q27* | q2* |
| 4 | q12 | q13* | q5 | q28 |
| 5 | q15* | q16 | q10* | q1 |
| 6 | q31* | q17 | q20 | q23 |
| 7 | q9* | q12 | q29 | q7 |
| 8 | q2* | q3 | q11 | q25 |
| 9 | q17 | q21 | q4 | q8 |
| 10 | q11 | q22 | q35 | q34 |
| 11 | q3 | | q33 | q24 |
| 12 | q27* | | q19 | |
| 13 | q16 | | q26 | |
| 14 | q6 | | | |
| 15 | q25 | | | |
| 16 | q20 | | | |
| 17 | q35 | | | |
| 18 | q5 | | | |
| 19 | q18 | | | |
| 20 | q1 | | | |
| 21 | q28 | | | |
| 22 | q24 | | | |
| 23 | q23 | | | |
| 24 | q19 | | | |
| 25 | q10* | | | |
| 26 | q22 | | | |
| 27 | q8 | | | |
| 28 | q29 | | | |
| 29 | q32 | | | |
| 30 | q4 | | | |
| 31 | q21 | | | |
| 32 | q7 | | | |
| 33 | q34 | | | |
| 34 | q26 | | | |
| 35 | q33 | | | |

α NIH-HEALS: NIH Healing Experience in All Life Stressors [6, 7].

*Selected questions for final 9-item short form.

Factor 1- Connection.

Factor 2 - Reflection and Introspection.

Factor 3 - Trust and Acceptance.

per the Department of Health and Human Services (HHS) guidelines which ensures compliance with 45 CFR part 46 and other regulatory requirements. All NIH IRBs are registered with the OHSRP at the HHS. The current study was approved by the OHSRP and was deemed IRB

exempt as data were collected deidentified and procedures only included paper-pencil questionnaires and were noninvasive. The requirement for signed written consent was waived by OHSRP; however, verbal consent was obtained from each participant prior to study procedures. Participation was voluntary and patients were informed that they could decline participation in the study.

## Results

To determine the candidate items for the short form, the NIH-HEALS questions were sorted by model selection and ranked considering both an overall assessment (full instrument) and categorization based on each specific factor: Connection (F1), Reflection and Introspection (F2), and Trust and Acceptance (F3) scores (Table 3). This process allowed us to identify which items make up the scale that are important for maintaining the same structure in the short form version as in the original version.

The questions underwent dual ranking, encompassing both an overall assessment from the full set of items and categorization by specific factors. Selection of questions was based on their placement among the top 25 out of 35 in the total ranking, as outlined in Table 4. Questions that did not perform well in the total ranking (e.g., q10) were subjected to further evaluation, incorporating factor ranking and clinical insights from experts. This evaluation process took into careful consideration the applicability of the scale and sought to preserve the original structural integrity of the questionnaire.

With the nine items already selected for the composition of the short form (Table 4), psychometric properties were evaluated in three different samples (n=164) to ascertain whether these selected items can accurately represent and maintain the structural composition of the full version.

First, the reliability of the 9-item NIH-HEALS short form in each sample subset resulted in Cronbach's alpha coefficients of 0.86 for Sample 1, 0.83 for Sample 2, 0.76 for Sample 3, and 0.84 for Sample 4; these results were similar to the reliability observed in the full NIH-HEALS 35-item (Chronbach's alpha = 0.89). Split-half reliability for the 9-item short form was also strong ($r_p$ = 0.81, $r_p$ = 0.80, $r_p$ = 0.73, and $r_p$ = 0.75 for Samples 1-4, respectively), slightly lower than the full scale ($r_p$ = 0.95). These results indicate a good internal consistency, confirming that the questions included in the short form are strongly correlated with each other. This, in

**Table 4. Selected questions for the 9-item NIH-HEALSα short form (SF) via LASSOβ analysis and qualitative assessment, by rank from overall full set of question items and from each instrument factor.**

| Factor | Question | | Overall Rank | Factor Rank |
|---|---|---|---|---|
| Connection | q15 | My personal religious practice is important to me. | 5 of 35 | 1 of 10 |
| | q14 | My religious beliefs help me feel calm when faced with difficult circumstances in life. | 1 of 35 | 2 of 10 |
| | q13 | My situation strengthened my connection to a higher power. | 2 of 35 | 4 of 10 |
| Reflection and Introspection | q31 | I have an increased sense of gratitude | 6 of 35 | 2 of 14 |
| | q9 | Working through thoughts about the possibility of dying brought meaning to my life. | 7 of 35 | 1 of 14 |
| | q10 | Difficult circumstances in my life have increased my compassion towards others. | 25 of 35 | 5 of 14 |
| | q27 | I take more time to be in the moment. | 12 of 35 | 3 of 14 |
| Trust and Acceptance | q30 | I have a sense of peace in my life. | 3 of 35 | 1 of 11 |
| | q2 | I have a sense of purpose in my life. | 8 of 35 | 3 of 11 |

α NIH-HEALS: NIH Healing Experience in All Life Stressors [6,7].

β Least Absolute Shrinkage and Selection Operator (LASSO) regression model

turn, suggests a reliable and consistent measure of the construct that the instrument aims to assess.

The next step in analyzing the psychometric properties involved evaluating the correlation between the 9-item short form and the original 35-item version. Results showed Pearson's correlation coefficients of 0.95 for Sample 1, 0.92 for Sample 2, 0.92 for Sample 3, and 0.96 for Sample 4 (p < 0.001 for each), revealing a highly positive association between the proposed instruments (Fig 1).

After the process of developing and analyzing the psychometric properties of the short form version, the NIH-HEALS-SF (Fig 2) is presented below:

## Discussion

The present article reports on the development of the NIH-HEALS-SF and the assessment of its psychometric properties. Constructing a brief measurement tool poses the challenge of finding a balance between the quantity of questionnaire items and other crucial factors, including the comprehensiveness of content and the statistical precision of scores.

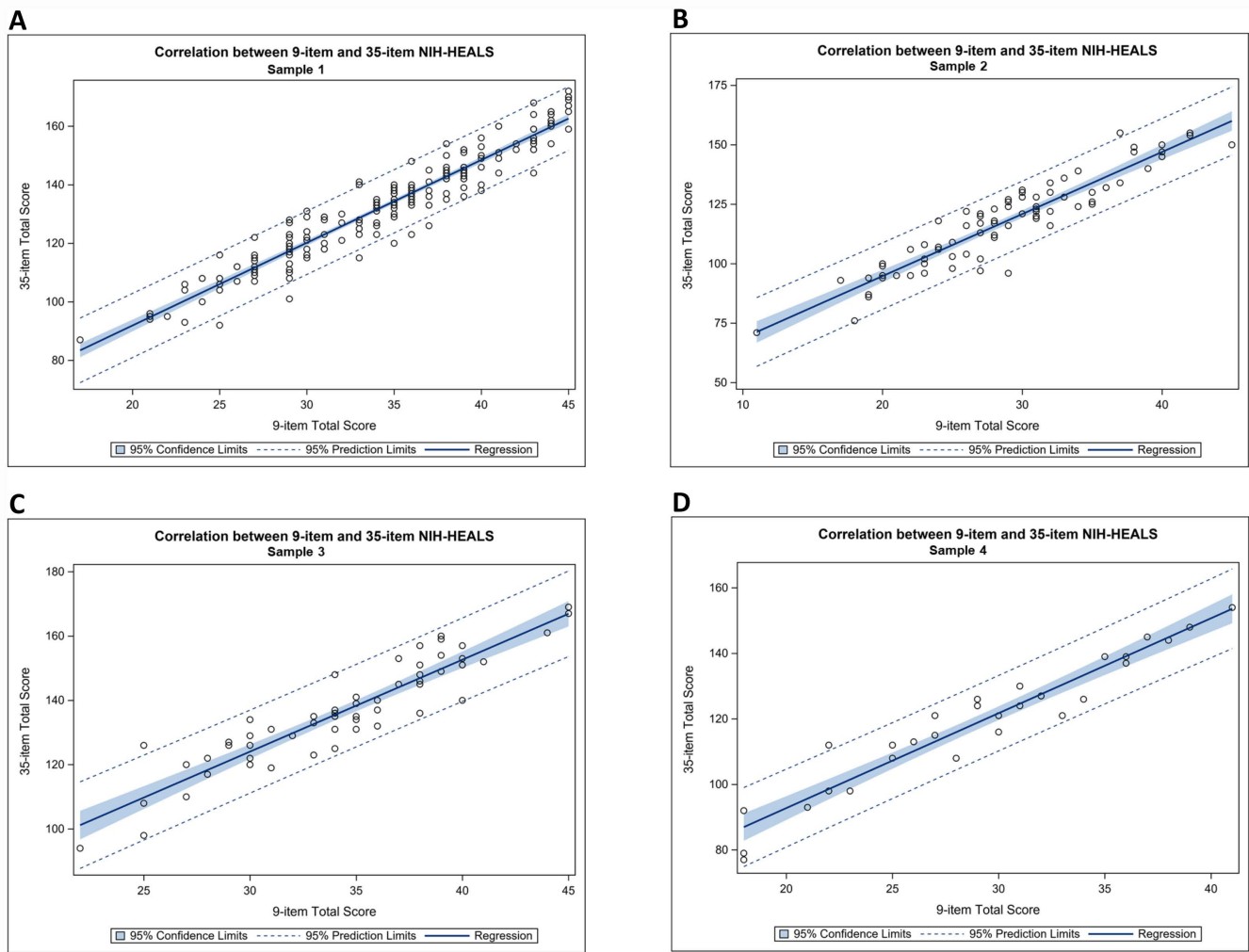

**Fig 1. Correlations between 9-item NIH-HEALS-SF and 35-item NIH-HEALS for each sample.**

**National Institute of Health
Healing Experience of All Life Stressors Short Form
(NIH-HEALS-SF)**

Please indicate how much you agree or disagree with each statement below.

| | Strongly Disagree | Disagree | Neither Agree or Disagree | Agree | Strongly Agree |
|---|---|---|---|---|---|
| I have a sense of purpose in my life. | 1 | 2 | 3 | 4 | 5 |
| Working through thoughts about the possibility of dying brings meaning to my life. | 1 | 2 | 3 | 4 | 5 |
| Difficult circumstances in my life have increased my compassion towards others. | 1 | 2 | 3 | 4 | 5 |
| Difficult situations strengthen my connection to a higher power. | 1 | 2 | 3 | 4 | 5 |
| My religious beliefs help me feel calm when faced with difficult circumstances in life. | 1 | 2 | 3 | 4 | 5 |
| My personal religious practice is important to me. | 1 | 2 | 3 | 4 | 5 |
| I take more time to be in the moment. | 1 | 2 | 3 | 4 | 5 |
| I have a sense of peace in my life. | 1 | 2 | 3 | 4 | 5 |
| I have an increased sense of gratitude. | 1 | 2 | 3 | 4 | 5 |
| Total score | | | | | |

**Fig 2. NIH-HEALS-SF measure (final version).**

LASSO analysis [16–18] was employed for the item selection. The statistical technique LASSO is considered to be a sound method for the development of health questionnaires short forms [19–23]. It can identify the most significant and representative items to maintain the structure of the full version of the instrument. LASSO provides an effective approach to balance the number of items, ensuring that the short form preserves the integrity of the assessed construct while reducing the burden on respondents [16–18]. Through this process, the original 35-item set was reduced to a 9-item set.

After selecting the strongest items (9) to compose the short-form version, NIH-HEALS-SF was applied to three different population samples for psychometric testing. The scores demonstrated an excellent internal consistency for Samples 1, 2, and 4 (Cronbach's alpha: 0.82-0.85), which is quite similar to that of the NIH-HEALS 35-item version (Cronbach's alpha: 0.89). In Sample 3, the Cronbach's alpha coefficient was 0.75, slightly lower than in the other samples. One possible reason for this discrepancy could be that this sample includes not only patients with gastric cancer but also family members that carry the gene for hereditary gastric cancer but are not ill at the present.

The strong correlation observed between the total scores of NIH-HEALS-SF underscores the consistency and coherence between the reduced set of items in NIH-HEALS-SF and the comprehensive 35-item version, affirming that the short form accurately captures the underlying constructs of psychosocial-spiritual well-being. This finding reinforces its potential as a valuable tool for practitioners and researchers seeking a more streamlined yet psychometrically sound measure of healing experiences in diverse populations.

As limitations of the study, external validation of the short-form version was not conducted, and should be evaluated in future studies and different populations. In addition, loss of information from eliminated items may result in missed insights about the populations under study. It is not known how the NIH-HEALS-SF may perform over time or whether it is equipped to detect longitudinal changes. Overall, the current results indicate that the NIH-HEALS-SF reflects a concise and comprehensive instrument aimed at optimizing the applicability of the tool without compromising the quality of assessment. This promising development suggests

potential practical applications for both the scientific and professional communities, paving the way for future investigations in evaluation of psycho-social-spiritual well-being.

## Acknowledgments

This research was supported by the Intramural Research Program of the NIH, Clinical Center. We would like to thank the patients and staff of the NIH Pain and Palliative Care Service for making this study possible.

## Author contributions

**Conceptualization:** Marcelli Cristine Vocci, Rezvan Ameli, Ann Berger.

**Data curation:** Ninet Sinaii.

**Formal analysis:** Ninet Sinaii.

**Funding acquisition:** Ann Berger.

**Investigation:** Marcelli Cristine Vocci, Polycarpe Bagereka, Rezvan Ameli.

**Methodology:** Marcelli Cristine Vocci, Rezvan Ameli, Ninet Sinaii, Ann Berger.

**Project administration:** Ann Berger.

**Software:** Ninet Sinaii.

**Supervision:** Ann Berger.

**Writing – original draft:** Marcelli Cristine Vocci.

**Writing – review & editing:** Polycarpe Bagereka, Rezvan Ameli, Ninet Sinaii, Jeremy L. Davis, Manish Agrawal, Ann Berger.

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
