## [Decision Letter · Decision Letter 0]

18 Nov 2024

PMEN-D-24-00117

Development of the National Institute of Health Healing Experience of All Life Stressors Short Form (NIH-HEALS-SF)

PLOS Mental Health

Dear Dr. Berger,

Thank you for submitting your manuscript to PLOS Mental Health and we apologise for the very severe delay - we appreciate that this would have been incredibly frustrating for you. After careful consideration of the reviewer comments, we feel that it has merit but does not fully meet PLOS Mental Health’s publication criteria as it currently stands. Therefore, we invite you to submit a revised version of the manuscript that addresses the points raised during the review process.

Please fully address the reviewer comments, which you can find at the end of this email and in the attachment.

We look forward to receiving your revised manuscript and please accept my apologies again.

Kind regards,

Karli Montague-Cardoso

Executive Editor

PLOS Mental Health

Additional Editor Comments (if provided):

Reviewers' comments:

Reviewer's Responses to Questions

**Comments to the Author**

1. Does this manuscript meet PLOS Mental Health’s publication criteria ? Is the manuscript technically sound, and do the data support the conclusions? The manuscript must describe methodologically and ethically rigorous research with conclusions that are appropriately drawn based on the data presented.

Reviewer #1: Yes

Reviewer #2: Partly

2. Has the statistical analysis been performed appropriately and rigorously?

Reviewer #1: No

Reviewer #2: I don't know

3. Have the authors made all data underlying the findings in their manuscript fully available (please refer to the Data Availability Statement at the start of the manuscript PDF file)?

Reviewer #1: Yes

Reviewer #2: Yes

4. Is the manuscript presented in an intelligible fashion and written in standard English?

Reviewer #1: Yes

Reviewer #2: Yes

5. Review Comments to the Author

Reviewer #1: Please review the sampling procedure and give rational based on scientific procedure to make better

need better interpretation of table and follow statistical norms for analysis

it it better to describe the information following the scientific procedure in methodology section.

Reviewer #2: This manuscript addresses the development of the NIH-HEALS short form (NIH-HEALS-SF), a tool aimed at reducing patient burden while maintaining the robust psychometric properties of the original scale. The topic is highly relevant, as the optimization of psychosocial-spiritual assessment tools enhances their usability in both clinical and research settings. The methodology is appropriate for scale reduction, and the results are compelling, demonstrating that the shortened version retains strong reliability and validity. However, there are areas where the manuscript could benefit from further clarification and elaboration to strengthen its impact.

1. The use of LASSO regression for item reduction is a strength of the study. However, the manuscript would benefit from additional explanation of why this method was chosen over alternative item selection techniques, such as exploratory or confirmatory factor analysis. Highlighting the advantages of LASSO in this context would strengthen the rationale for the methodology. Why not using an EFA approach? Another piece of information is missing here is what statistical software/package is used here for conducting LASSO and the model fit indices were not reported here either.

I think more elaboration of this method and statistical output is needed for readers to better understand the process of item selection.

2. It is unclear how the three factors (Connection, Reflection and Introspection, and Trust and Acceptance) guided the final selection of items. Were these factors weighted equally in the selection process? Including a brief discussion of the factor-specific contribution to the short form would provide greater transparency.

3. The study mentions validation using data from three distinct samples. I think a discussion of whether the psychometric properties of the short form varied across subgroups (e.g., different clinical populations) would enhance the manuscript. Are there signficant differences in the level of healing across different samples or populations.

4. The reported Cronbach's alpha values (0.75–0.85) indicate good internal consistency, but the manuscript could elaborate on how this compares to the original scale. Is there any trade-off in reliability due to the reduction in items?

5. It would also be helpful to discuss any potential limitations of using the short form, such as the potential loss of nuanced insights or sensitivity in detecting changes over time.

6. PLOS authors have the option to publish the peer review history of their article (what does this mean? ). If published, this will include your full peer review and any attached files.

**Do you want your identity to be public for this peer review?** For information about this choice, including consent withdrawal, please see our Privacy Policy .

Reviewer #1: No

Reviewer #2: No

---

## [Editor Report · Decision Letter 1]

7 Feb 2025

Development of the National Institute of Health Healing Experience of All Life Stressors Short Form (NIH-HEALS-SF)

PMEN-D-24-00117R1

Dear Dr. Berger,

We are pleased to inform you that your manuscript 'Development of the National Institute of Health Healing Experience of All Life Stressors Short Form (NIH-HEALS-SF)' has been provisionally accepted for publication in PLOS Mental Health.

Best regards,

Karli Montague-Cardoso

Executive Editor

PLOS Mental Health